# Barriers to equitable access to quality trauma care in Rwanda: a qualitative study

Pascal Nzasabimana,[1] Agnieszka Ignatowicz,[2] Barnabas Tobi Alayande [ID] ,[3,4] Abdul-Malik Abdul-Latif,[2] Maria Lisa Odland [ID] ,[2,5,6,7] Justine Davies,[2,8,9] Abebe Bekele,[3] Jean Claude Byiringiro [ID] [10]

American College of Surgeons, Clinical Congress, 2022

JD, AB and JCB are joint senior authors.

For numbered affiliations see end of article.

**Correspondence to**
Dr Agnieszka Ignatowicz;
a.m.ignatowicz@bham.ac.uk

## ABSTRACT

**Objectives** Using the 'Four Delay' framework, our study aimed to identify and explore barriers to accessing quality injury care from the injured patients', caregivers' and community leaders' perspectives.

**Design** A qualitative study assessing barriers to trauma care comprising 20 in-depth semistructured interviews and 4 focus group discussions was conducted. The data were analysed thematically.

**Setting** This qualitative study was conducted in Rwanda's rural Burera District, located in the Northern Province, and in Kigali City, the country's urban capital, to capture both the rural and urban population's experiences of being injured.

**Participants** Purposively selected participants were individuals from urban and rural communities who had accessed injury care in the previous 6 months or cared for the injured people, and community leaders. Fifty-one participants, 13 females and 38 males ranging from 21 to 68 years of age participated in interviews and focus group discussions. Thirty-six (71%) were former trauma patients with a wide range of injuries including fractured long bones (9, 45%), other fractures, head injury, polytrauma (3, 15% each), abdominal trauma (1, 5%), and lacerations (1, 5%), while the rest were caregivers and community leaders.

**Results** Multiple barriers were identified cutting across all levels of the 'Four Delays' framework, including barriers to seeking, reaching, receiving and remaining in care. Key barriers mentioned by participants in both interviews and focus group discussions were: lack of community health insurance, limited access to ambulances, insufficient number of trauma care specialists and a high volume of trauma patients. The rigid referral process and lack of decentralised rehabilitation services were also identified as significant barriers to accessing quality care for injured patients.

**Conclusions** Future interventions to improve access to injury care in Rwanda must be informed by the identified barriers along the spectrum of care, from the point of injury to receipt of care and rehabilitation.

## INTRODUCTION

Historically, global health initiatives have prioritised infectious diseases. However, in 2019 alone, the world recorded 4.3 million deaths from trauma, more deaths than malaria, tuberculosis and AIDS combined.[1]

## STRENGTHS AND LIMITATIONS OF THIS STUDY

⇒ The study assessed access to trauma care access using a comprehensive 'Four Delay' framework that also emphasised remaining in care.

⇒ A multistakeholder perspective on delay is obtained by triangulating the perspectives of the service user, caregiver and community leader.

⇒ Individual in-depth interviews and focus group discussions in the local language were carried out on a diverse group of participants from Rwanda's urban and rural communities.

⇒ All transcripts of interviews and focus group discussions were coded in parallel by a context and local language fluent member as well as qualitative data experts to ensure rigour.

⇒ Seventy-five per cent of study participants are male, and even though the majority of injured patients in Rwanda are male, we recognise that this may limit the generalisability of results.

This burden is heaviest in low-income and middle-income countries (LMICs), where injury rates are high, trauma systems are underdeveloped and patients requiring emergency care are young and economically active.[2] Up to 11% of deaths in Rwanda and 22% of deaths in the largest urban city of Kigali, housing 1 135 428 people according to the 2012 population census, are due to injury, with the most common cause being road traffic collisions.[3][4] Although injury prevention initiatives are necessary, identifying health system deficiencies and barriers that hinder access to safe, affordable, good quality and timely trauma care is key to reducing preventable deaths from people who are inevitably injured. A South African study suggests that strengthening health systems by focusing on interfacility transfer systems would reduce up to 59% of preventable trauma deaths resulting from delays in receiving care. In addition, the same study argues that 23% and 18% of deaths from injury can be prevented by mitigating delays in seeking and reaching trauma care, respectively.[5] Identifying and

characterising specific barriers to care is crucial for developing and implementing successful interventions, and reducing the burden of injury.[6]

The 'Three Delay' framework was developed originally to identify and mitigate barriers to maternal and child care but has been successfully applied to emergency trauma care and injury care.[7] A systematic review and evidence synthesis mapping this framework to injury health system assessments found that approximately 79% of studies focused on delays within facilities, 46% assessed delays in reaching care, and only 10% assessed delays in seeking care.[8] Research from sub-Saharan Africa comprised 40.1% of the studies assessed in this systematic review. A more comprehensive 'Four Delay' framework for assessing trauma care access has been suggested, comprising delays in seeking, reaching, receiving and remaining in care.[9] This more comprehensive model emphasises the need for holistic care, postdischarge physiotherapy and other rehabilitation for the injured patient. In a workshop carried out to map barriers to trauma care in Rwanda using the four-delay framework, health providers, academia and government representatives identified 19 high-priority barriers faced by trauma patients and prioritised the top four. Three of the prioritised four were related to receiving care, but most of the barriers cut across multiple components of the four-delay model.[9] However, the perspectives of the injured patients in the Rwandan context have not yet been explored. Our study focused on identifying and exploring barriers to accessing quality care from the perspective of the injured patients to develop priorities for trauma system interventions in Rwanda.

## METHODS

This study was conducted using a grounded theory qualitative approach and a constructivist research paradigm. The Standards for Reporting Qualitative Research checklist guided all components of the writing and reporting of this study. This study was conducted concurrently with studies in Ghana and South Africa as part of the equitable access to quality trauma systems in lower and middle-income countries: a qualitative study assessing gaps and developing priorities.[10]

### Patient and public involvement statement

This study was preceded by a National Trauma Symposium in 2019 organised by the Rwanda Ministry of Health and partners, where over 100 community stakeholders including representatives of the health ministry, Rwanda National Police, local non-governmental organisations, universities, emergency respondents, students and medical practitioners identified access to trauma care as a national research priority.[10] This work was also premised on a 2-day workshop held in Kigali, Rwanda in May 2019 with representation from community, academia and government which aimed to map barriers to accessing care from injury to rehabilitation and is in line with public

priorities.[11 12] Injured patients, caregivers and other stakeholders were involved in the design and execution of the study.

### Researcher characteristics and reflexivity

Four of the eight authors live and work in Rwanda and served as context experts. Of these four, three are surgical providers who provide various degrees of trauma care and leadership to the trauma system. Overall leadership for the work in Rwanda was provided by a Rwandan trauma surgeon. Interviews were conducted by an investigator who is from rural Rwanda, speaks Kinyarwanda and has lived experience of the health system and delays. There was adequate gender balance on the research team. Two UK-based investigators are qualitative method experts. The team is experienced in such studies as seven of the eight investigators were involved in similar surgical systems assessments in Ghana and South Africa.

### Context

The study was conducted in Rwanda's rural Burera District, located in the Northern Province, and in Kigali City, the country's urban capital. Kigali City has a population of approximately 1 135 428 people concentrated on 1760/km², whereas Burera district has a population of 336 455 people distributed over 522/km².[11] Burera, in terms of population and infrastructure, is a typical rural Rwandan district. Rwanda's urbanisation is dominated by Kigali, which houses nearly 60% of the urban population and continues to experience rapid population and economic growth while modernising and upgrading its infrastructure and transportation systems.[12] Road traffic injuries are a major cause of trauma deaths in Rwanda, particularly among young men.[13] Injury care pathways generally start with community health workers who are embedded in the communities, who refer to local health centres, and subsequently to second tier district hospitals, who refer to tertiary teaching hospitals and eventually to the country's single quaternary health institution. Many health facilities have ambulances that can transport injured patients to higher levels of care. The country's terrain is largely hilly. Most Rwandans (over 90%) are covered by *Mutuelle de Santé*, Rwanda's elaborated Community-Based Health Insurance program.

### Sampling strategy

Using purposive sampling, participants of the 20 in-depth interviews and 2 focus group discussions were drawn from the hospital register if they were over the age of 18 and had either received injury care themselves within the previous 6 months or had cared for trauma victims who had received care. Local religious leaders, business leaders, service provider leadership, traditional healers and local government authority figures participated in two community leaders' focus group discussions. Individuals under the age of 18 or with any intellectual impairments or cognitive deficit that would limit informed consent and participation, as well as service users accessing acute

care during the study period, were excluded. Individuals who returned for follow-up and successfully remained in care however were not excluded.

## Data collection methods

A list of patients treated for traumatic injuries in the last 6 months was obtained using information obtained from hospital admission registers at the emergency department and surgical wards of Centre Hospitalier Universitaire de Kigali (CHUK), Rwanda's leading centre for tertiary trauma care provision, and Butaro District Hospital, a district facility in northern Rwanda. Patient caregivers identified through the trauma patients also took part in the study. A list of community leaders within the facilities' catchment area was also obtained through the administration departments of both hospitals.

Participants were oriented to the study by the interviewer (PN) who explained the aims and objectives of the study to participants in Kinyarwanda, ensuring verbal feedback from potential participants to confirm that they understood the study, and their potential role. Written informed consent was subsequently obtained. Individual interviews lasted 30 min on average, and focus groups discussions lasted between one and one and a half hours for six to nine participants.

## Data collection instruments and analysis

Data collection was guided by study instruments designed by qualitative experts in English, which were translated to Kinyarwanda (see online supplemental file 1). Interviewers used an audio recorder to record the semistructured interviews and focus group discussions. Recordings were transcribed, and translated into English before being uploaded to QSR International's NVivo V.12 qualitative data analysis software for analysis.[14] Thematic analysis of data was performed, as well as inductive and deductive coding. Themes were developed iteratively with reference to the Four Delay framework and according to whether they represent the delay from seeking, reaching, receiving or remaining in care.[9] We defined the first delay (delay in seeking care) as delays from the time the injury occurred to the time a decision was made to seek medical care. The second delay (delay in reaching care) referred to delays from the time a decision was made to seek medical care to the time when an appropriate medical facility capable of managing the particular injury (prior to registration) was accessed. The third delay (delay in receiving care) was defined as delays beginning from the point of entering a capable facility to the point of receiving specific management for the injury. This could include damage control interventions for the injury initiated on admission to a hospital intended to manage an injured patient's condition or injury. This specific management can be operative surgical care when there is the indication for surgery, but is often non-operative, like the application of casts, administration of medications or admission for observation. The fourth delay (delays in remaining in care) was defined as delays from the point of having received definitive care to the point of final discharge from follow-up. Remaining in care encompassed the time periods of continued in-hospital care beyond definitive care, discharge and physical therapy.[9] We defined injury as wound or a condition of the body caused by external force or exchange of energy between the body and the environment of such magnitude that is beyond the resilience of the body.

To ensure rigour, all transcripts of interviews and focus group discussions were coded in parallel by a context and local language fluent member (PN) and qualitative data experts (AI and MLO). The analysis team met on a regular basis to discuss the coding process. Any conflicts that arose during the independent coding process were resolved by group consensus. The qualitative analysis procedure was the same for interviews and focus groups. Following coding and the identification of initial categories, data from interviews and focus groups were combined. The final list of themes was reviewed and agreed on by the entire investigator team. Barriers were included based on single mention.

## RESULTS

A total of 51 participants, 28 from the rural Burera District and 23 from the urban Kigali city, participated in the interviews and focus group discussions. Of all the participants, 13 were female and 38 were male, ranging from 21 to 68 years of age. Thirty-six (71%) of the participants were patients (service users), while the rest were caregivers and community leaders. For the individual in-depth interviews, 10 patients (50%) were from urban Kigali, and 10 (50%) from rural Burera. They presented with a wide range of injuries including fractured long bones (9, 45%), other fractures (3, 15%), head injury (3, 15%), polytrauma (3, 15%), abdominal trauma (1, 5%) and lacerations (1, 5%). The major mechanism of injury was involvement in road traffic collisions in the urban setting (9, 90%), and less so in the rural settings (5, 50%). There were more falls from heights in the hilly rural settings (4, 40%), than in the urban areas. Only one rural report of assault was captured.

Sixty barriers were identified across all four delays. The most mentioned barriers were the insufficient number of competent trauma care providers, the rigid referral process and facility-related delays in delivery of care. Box 1 shows the barriers to injury care identified by participants across all delays. Figure 1 shows a word cloud that pictorially represents the barriers to injury care based on mentions in the individual in-depth interviews.

## Delays in seeking care

A typical rural road accident scene would be crowded with bystanders observing the injured and discussing their presumed prognosis. Others at the site of injury would attempt to provide first aid, call the ambulance service and the police, reach out to any available family members using the injured individual's cell phone, manage traffic movements at the crash site, and obtain any information

## Box 1  Barriers to injury care identified by participants across all delays

**First delays: delays in seeking care**
⇒ Community health workforce related delays: perceived sufficiency of respondent's interventions.
⇒ Community leadership related delays: community conflict management process.
⇒ Health financing related delays: preinjury financial challenges and limited comprehensiveness of the insurance system.
⇒ Health services delivery related delays: perceived inaccessibility of the 'right' hospital or acute care facility, lack of awareness of healthcare and injury care systems, lack of confidence in healthcare system and the reference for alternative healthcare.
⇒ Knowledge on health information related delays: underestimation of the severity of the injury, religious beliefs, respondent rumours about seeking injury care, shared decision-making (contribution of community health workers and village leaders).

**Second delays: delays in reaching care**
⇒ Healthcare services access related delays: limited geographical coverage of ambulance services, cost of transport to hospital and between hospitals, unavailability of transportation, limited access to ambulances, ignorance of ambulance call process and emergency numbers, high urban road traffic density and time of the day of injury.
⇒ Infrastructure-related delays: poor road infrastructure.
⇒ Health financing related delays: fear of spending on private transport cost and absence of personal health insurance.
⇒ Community leadership related delays: unwillingness of passers-by to assist, fear of legal complications for intervening private car owners, community organisation required for reaching facilities, lack of social assistance and support, and accident investigation process by the police.
⇒ Community health workforce related delays: delays in arrival of emergency respondents, occurrence of multiple road traffic accidents, poorly coordinated interfacility transfers.

**Third delays: delays in receiving care**
⇒ Health financing related delays: lack of financial means, absence of health insurance, cumbersome facility invoicing and cost recovery system, unavailability of some community health insurance staff during work hours, unaffordable costs of materials and medications purchased in non-government funded facilities, and the inability to use community health insurance in private pharmacies.
⇒ Health services delivery related delays: high patient load and limited capacity of facilities, inadequate facility infrastructure, inadequate follow-up of privately referred patients to the point of receiving care by referring physicians, and the reduced staff after daylight working hours.
⇒ Health workforce related delays: insufficient skilled medical and specialist staff, unavailability of certain specialised services and the high cost of these services outside the country, poorly organised interfacility referral system, and traditional healers and their interface with the health system.
⇒ Health information related delays: difficulty in the use of electronic medical systems by healthcare staff.
⇒ Medical products related delays: unavailable equipment within facilities, and unavailable medications or other treatment within facilities.

**Fourth delays: delays in remaining care**

*Continued*

## Box 1  Continued

⇒ Health services delivery related delays: poor counselling of patient in need for rehabilitation by healthcare workers, previous unsatisfactory experience of health facility care, perceived early discharge in an unsatisfactory health condition, complications of alternative home-based care which may limit patients' movement, limited opening hours of some facilities, cumbersome referral system for gaining appointments and multiple transfers required for community health insurance users, lack of triage services at receptions; slow service delivery at health centres, and difficult follow-up process for discharged patients.
⇒ Health workforce related delays: traditional healers counselling patients against returning for follow-up, multitasking by healthcare workers and lack of staff supervision and many responsibilities on one staff in health centres, and insufficiency of staff and decrease in confidence at the health centre.
⇒ Health information related delays: use of paper-based health records.
⇒ Transport-related delays: lack of transportation for follow-up, and difficult terrain between home and location for rehabilitation or physiotherapy.

required by the police for postaccident legal procedures. The decision to move victims to a nearby health facility would usually be made after some (usually intense) discussion among bystanders. In the rural setting, the most proximate facilities are typically health centres and health posts and participants mentioned that only a small proportion of the injured would delay seeking care and return home for self-management or seek the services of a traditional healer.

At the scene of an accident, it was reported that in most cases, the initial decision to seek care was deferred by victims or bystanders on account of distance and time to health facilities from the accident location, lack of financial means including considerations of whether or not injured individuals had community health insurance and reticence about going through the process of activating community health insurance payments.

> (…) when the patient doesn't have health insurance and doesn't have money, they fear going to health facilities as it is complicated. In Kigali city, it is even more complicated as you may go to the hospital and they ask you to pay around 50 to 60 thousand, and it is complicated. Then people may miss going to health facilities due to that. (FGDPatUrban01)

Furthermore, a poor understanding of the trauma victim's health condition and the need for care, coupled with an inaccurate perception of the severity of injury by the individual and bystanders, often delayed the decision to seek care. Occasionally, community health workers and village leaders would contribute to this delay if they felt the injury was not severe enough to necessitate care.

> … (…) a case of a mother who had a child who was kicked against the wall by the sheep, and they said it was not serious, saying that it was normal for the

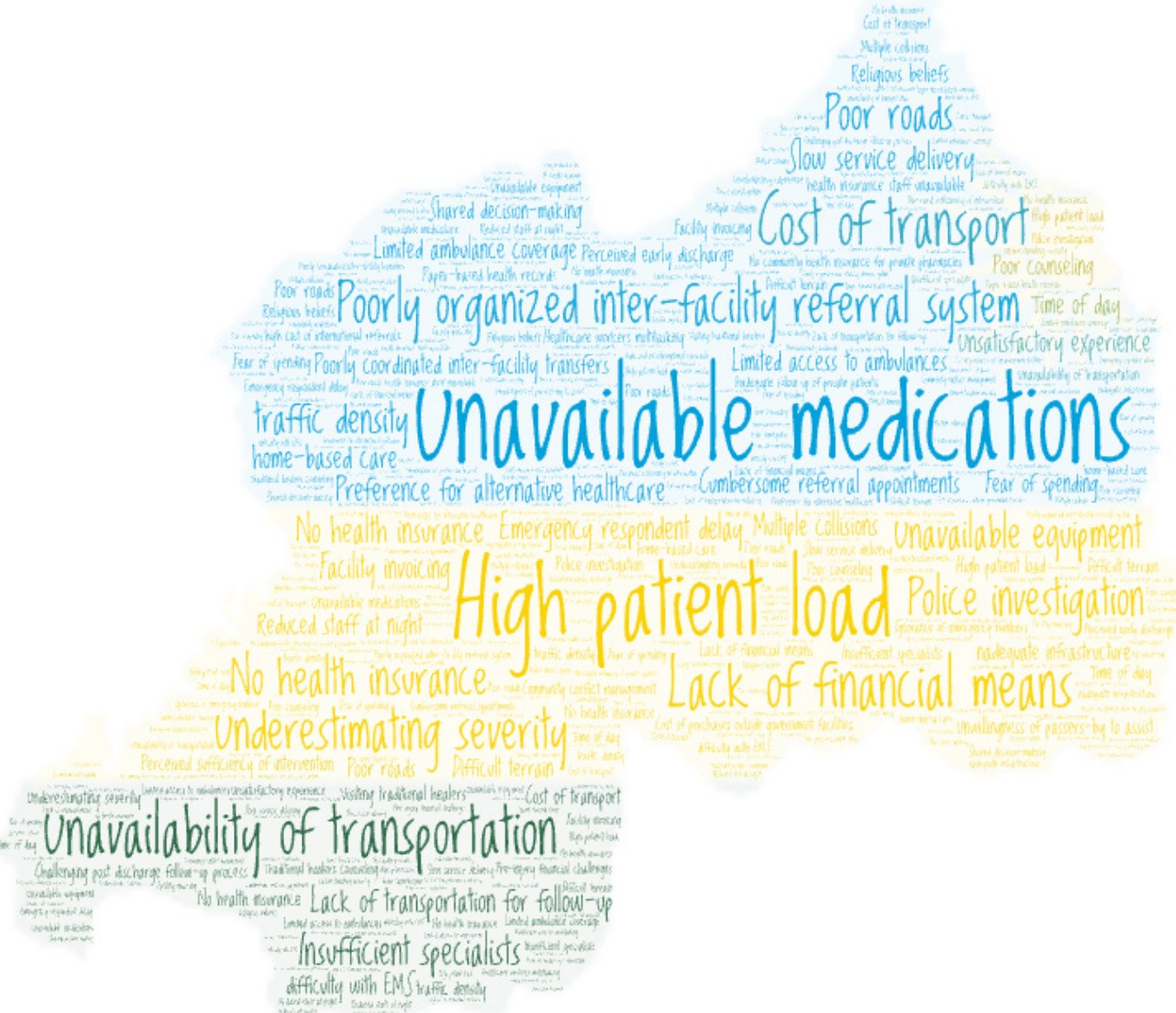

**Figure 1** Word cloud of barriers to injury care in Rwanda.

children to fall and have minor pain, until the child continued to swell to the level that they decided to take him to the hospital very late. (FGDPatRural01)

Sociocultural and religious beliefs opposed to orthodox healthcare and prior positive experiences with traditional medicine use also contributed to this delay.

What I also observed is that an individual may have an injury causing bleeding, and instead of taking him to the hospital, they choose to apply herbal medications and later go to the hospital with complications. (FGDPatRural01). People's belief systems and confidence in traditional medicine contribute a lot to the decisions they take on what to do after having an accident. (FGDComUrban01)

In rural Rwanda, it is the community and family that often decide whether or not an injured member should receive care at a healthcare facility. This shared decision-making process may further delay seeking care. In cases of violent communal conflict, the community conflict management process also delays the decision to seek care, as settling the conflict is prioritised over taking decisions to take victims to a care facility.

In the community (…) the members of the wider community first come to see your husband or wife and the family then collectively decide to take you to the health facility. (FGDComRural01)

### Delays in reaching care

The injured individuals would usually be transported in traditional stretchers and on motorbikes, depending on their condition. In contrast, injured in urban Kigali would be typically carried by emergency services

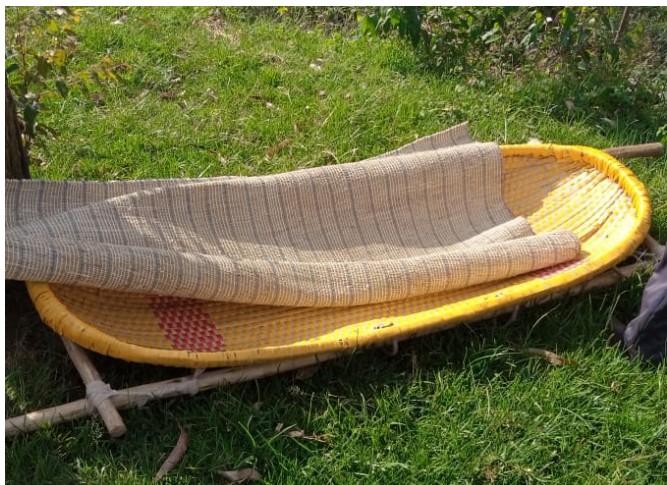

**Figure 2** A Rwandan traditional stretcher. Photo by Barnabas Alayande.

ambulances to district and referral hospitals within the city.

In the rural setting, they reported using different modes of transportation from accident scenes to nearby health facilities. This ranged from walking on foot, riding commercial motorcycles, using traditional stretchers (see figure 2), ambulances and renting private cars. The most common means were commercial motorcycles, if the condition of the injured permitted, and traditional stretchers carried by four members of the community were reserved for severely injured individuals in the rural setting.

Only a few of the injured people reported reaching the health facilities by ambulances. The perception was that ambulances were scarce and were only used to transport pregnant women and children from home to hospital. Furthermore, in the rural areas, most people believed that the ambulance were only used for interfacility transfer and not for initial transport to the hospital.

I know that you have to first take your patient to the primary health facility, and if they decide to refer him, it is then the time they can call an ambulance. (FGDPatRural01)

This was in contrast to urban participants who reported the use of ambulances as the primary means to reach health facility. Alternatives in Kigali were only used when ambulances were not required, available or accessible.

… I remember the last time an accident occurred, and there were so many accidents (…) Then there was the surprise of how they instructed the ambulance to deviate its initial direction to pass there and bring one medical personnel at the accident scene to return later after taking that first patient to the hospital. (FGDComUrban01)

Furthermore, the cost of private transport and the unwillingness of private drivers to assist in transporting the injured contributed to the delay in reaching the care.

Private car owners, in particular, if they experienced complicated reception processes at health facilities in the past, were not willing to help transport the patients

(…) apart from having a human heart, people are hesitant to help in that situation because bringing the accident victim to the medical facility suggests that you are willing to spend your time there. (FGDComUrban01)

Additionally, high traffic density during rush hour in Kigali, the occurrence of simultaneous road traffic accidents and COVID-19 curfew-related delays were cited.

The most challenge is traffic jam, the ambulance may come on time but the road is busy with traffic jam. (FGDPatUrban01) and during COVID-19, when there were curfew hours and you wanted to reach the hospital, the motorcycle rider would ask you for a lot of money, saying that he would face the traffic police and that he would need to use shortcut roads which are not safe. If you were lucky, he would take you to the hospital (…) [but] you had to pay a lot of money, almost three times the normal cost since you had no alternative. (FGDPatRural01)

The absence of personal insurance also led to delays in reaching care.

… [a] patient who has personal financial challenges (…) advised me to first solve issues of community health insurance before going to the hospital to get care. (PatRural002)

The interfacility referral and transfer process also contributed to delays in receiving care, especially when it came to asking trauma victims to reach a higher level facility unassisted and not in the ambulance.

From the health center, they gave us a referral to [name of district hospital] without facility transport. Here, they also provided us with basic healthcare and a transfer to Ruhengeri hospital [referral hospital], but we had to do it ourselves. We needed four people to lift her [the injured patient] as she is old; it required me to take four motorcycles, one for her, one for me, and others for the ladies to care for her on each trip (FGDPatRural01)

### Delays in receiving care

Most of the participants in rural Burera first received care at the health post or the health centre level, while those in Kigali City who used mostly emergency ambulance services to reach care, commonly accessed higher levels of care through district hospitals and the University Teaching Hospital of Kigali (CHUK). However, across the board, the required medication and diagnostic equipment was not always available at these health facilities, and the prohibitive cost of trauma care materials and medications purchased outside of the government facilities delayed receiving care. Facility overload also emerged

as a barrier, as hospitals had limited capacity to attend to several patients at once.

> (…) patients with multiple fractures come in and are laid on the ground. The hospital has a serious problem of shortage of beds and they do not have space to put all emergencies they receive. (FGDPatUrban01). (…) the infrastructure we have doesn't meet the need as you may find, even the doctors are few. Patients they have to operate on don't have enough space for post-operative care, and this may limit services. (FGDComUrban01)

Some participants experienced delays in receiving care within the facility because of an insufficient number of skilled trauma care providers or specialists. One of the patients interviewed talked about experiencing a delay in receiving care because healthcare providers did not recognise the severity of their injuries.

> (…) they have low skills (…) like in case of fracture. You see, a general physician may not have enough skills on it, and you see them discussing it, and you hear them. They delayed doing this and you had to do this; you know, medical staff, the one who came first [to see me] was clearly working in slow-motion for unjustified reasons. (PatRural004)

Several participants mentioned the organisation and flow of patients in the health facility process as being responsible for delays in receiving care. Specific barriers included gaps in the electronic medical records, a lengthy invoicing and record recovery system; difficulties in effecting payments; inadequate communication by healthcare providers coupled with the patient's lack of familiarity with the health facility and long or short duration of treatment.

> The patient should come and not be required to wait on those benches. They should go immediately to the cashier to pay and see their doctor. It would be better than sitting on those benches, getting a number before going on to pay. Furthermore, some people are assisted by security staff during the payment procedure. Even if you come first, you run the risk of going home unattended by the doctor, as they assist many people. (FGDPatUrban01)

Another patient said "I was discharged on Friday, but the process could not be completed that day, and some services were unavailable on Saturday and Sunday, so I had to wait until Monday. I believe they should create a shortcut so that the person who discharge you can use their technology to show the credit you have and pay directly." (PatUrban002)

Additionally, "People have a problem when they discharge them and no longer consider their situation…. you observe for a few days, they discharge you. They released me while both of my legs were stiff from the prolonged suspension." (FGDPatUrban01)

Furthermore, issues surrounding insurance were cited by participants as causing delays at the facility. Patients with community health insurance reported that insurance staff were not always available to attend to patients during work hours, especially at the primary level of care.

> (…) I saw it in two facilities. You meet him [the insurance staff], and a short while later, when you return to see him, he is not at his workstation, and you have to wait while you are severely suffering. It is mandatory to see him as you cannot be attended to if the insurance worker does not help you. (PatUrban001)

Within public facilities, uninsured patients and those without financial means to purchase prescribed medication faced delays resulting from a relatively high direct cost of care and a long payment process that involved seeking hospital social and administrative approval to secure prescribed medication on credit. Outside government facilities, such as community pharmacies, health insurance was not accepted and the high cost of purchasing medications out of packet was a barrier to receiving care.

> If you have an accident and they have to put in the left leg prosthesis and they don't have it here, they send you to buy it for around 50,000 to 100,000 Rwandan Francs, while if you find it here, you pay only 10,000 Rwandan Francs [with community health insurance]. (FGDPatUrban01). Whenever they send you outside [of the health facility to private pharmacies], regardless of your insurance status, the amount of money billed for medication will be too much for you (FGDPatUrban01)

### Delays to remaining in care

Most participants felt injured patients had been discharged before full recovery due to insufficient beds and the need to admit other trauma patients. This led to difficulties in returning for follow-up as the patient's condition was not optimal for the strain of the travel and the visit.

> Often, the patient is discharged before achieving total recovery (…) the doctor decides to continue following up with the patient while at home. In that situation, the patient may not be satisfied. (FGDComUrban01)

> (…) the long travel on foot made it swollen, and I reached there with the arm all swollen. I had to be there early, an hour and a half before any physical examination, so that the swelling would go down and the doctor would not be confused. I don't want him to think that I have another problem while the reason for the swelling was actually from traveling for the appointment. (PatUrban010)

While agreeing that they had been given follow-up and rehabilitation appointments, the distance from home to rehabilitation facilities, difficulties in arranging transport and the cost of transport were common barriers

to returning. Some injured patients visited traditional healers and were subsequently dissuaded from attending the follow-up appointments in the health facility or hospital.

Most of the interviewed participants were community health insurance users. They identified delays caused by the need for patients to present a new transfer from lower levels of care before being attended to for follow-up at higher levels of care from which they were discharged. Presenting for rehabilitation or follow-up at a tertiary facility necessitates returning to the health centre and district hospital to ensure insurance claim payment by obtaining a new referral or transfer for each visit that exceeds 30 days following discharge.

> When I was discharged, I came back [for my follow-up appointment] and stayed there till 3:00 pm. When I finally reached those cashiers, they asked me for the transfer, and I said I had an appointment. They insisted I needed the transfers. I asked them why because I was previously admitted, and they told me to go back to the health center and then to the district hospital to get the transfers and to reschedule my appointment for later. I (…) finally went back home, then to the health center and from there to the district hospital before finally coming back again to ask for a new appointment. (FGDPatUrban01)

## DISCUSSION

This study is the first qualitative study, of which we are aware, to identify barriers to injury care from the point of injury to rehabilitation in both rural and urban settings in Rwanda, using the four delays framework. A recent systematic review found that very few studies in lower and middle-income countries have comprehensively assessed barriers to access to quality trauma care[8] from being injured to being discharged from care. What is more, we focused on urban and rural settings in recognition of possible disparities that may exist in accessing injury care in Rwanda. Multiple barriers to injury care were identified across the four delays, in both rural and urban settings. The barriers most mentioned by patients and community leaders were 'insufficient number of competent trauma care providers', 'rigid referral process' and 'facility-related delays in delivery of care'. These findings echo the results of a multidisciplinary stakeholder consultation aimed at mapping injury care access in Rwanda, which found that 'training and retention of specialist staff', 'geographical coverage of referral trauma centres' and 'lack of protocol for a bypass to referral centres' among the top 4 barriers identified by the participants.[9] The similarity of findings from these two studies, despite the differing perspectives of contributors, suggests that interventions aimed at ensuring access to safe, timely, affordable and quality injury care should take into consideration these specific barriers.

Regarding transport for injured patients to facilities that can manage their injuries, a limited number of ambulances available to convey injured patients from the point of injury to the appropriate trauma care facility was identified as a cause of delay. A systematic review of literature in LMICs found that often ambulances were not available, but even when ambulances are available, many emergency and trauma patients are forced to use alternative means of transport.[15] Rwanda is relatively unusual in sub-Saharan Africa in having a country-wide ambulance service. However, discussions at the aforementioned workshop and in the 2019 Rwanda trauma symposium highlighted that efficiency could be improved with better location of the patient and communication between the ambulances and facilities.[9 16] Studies are ongoing to address these issues.[17 18] Although our participants in Kigali stated that injured people often used ambulances, it was clear that their availability was much lower in rural areas. Rwanda's unique hilly topography must be considered when implementing low-cost and innovative ambulance services for rural populations. Another perspective, however, may be that participants did not know how to access the ambulance service. The widespread perception in this study was that in rural areas, ambulance use was restricted to only transporting pregnant women or sick children from home to hospital. Rwanda's Service d'Aide Medicale d'Urgence was founded in 2008 and now has up to 225 ambulances (5 per district). This results in an ambulance to population ratio of 1:48 000 and exceeds the WHO recommendation of 1:50 000.[15] The rural–urban disparity in the perception of the availability of ambulances for injury care should be addressed. Community education, alongside improved coordination and logistics of ambulance distribution for injury care, should to be optimised.[19 20]

In sub-Saharan Africa, surgical and anaesthetist densities are generally low. By 2015, Rwanda had only 0.79 per 100 000 skilled health workers, a significantly lower ratio than in high-income countries.[21] Trauma care providers are perceived to be in short supply and inequitably distributed in favour of urban areas. The participants in this study identified 'insufficient number of qualified trauma care providers' as a major barrier to receiving injury care. Hutch *et al* found that only 20 surgeons were trained by the College of Surgeons for East Central and Southern Africa in the 13 years leading to 2013.[22] Despite this low number of surgeons, Rwanda has a high level of surgeon retention (90%) following training.[22] However, this retention is particularly skewed towards urban areas, and an attrition rate of up to 80% for doctors in the rural setting has been identified.[23] Rwandan anesthesiologists and surgeons with specialised training work almost entirely in urban referral hospitals, leaving surgical care at rural district hospitals largely to general practice physicians and nurses.[24] The presence of trained anaesthetists and the availability of anaesthesia are essential to operative injury care. A 2017 study found that over a third of anaesthetists trained in Rwanda had emigrated for a variety of reasons,

ranging from a desire for better remuneration to a lack of equipment and medication for safe anaesthesia, occupational isolation in rural settings and demoralisation.[25] Only about 75 physiotherapists serve Rwanda's entire population of over 13 million people.[26] This issue of insufficient trauma care providers is in keeping with findings from other LMICs. In a systematic review of barriers to out-of-hospital emergency trauma care, over 60% of them identified a lack of skilled personnel as a key barrier[15]; lack of available, trained and motivated clinical staff was the second highest barrier to access to quality trauma care identified in an international Delphi exercise.[20] The WHO recognises the role of various levels of healthcare delivery in trauma management. Non-physician providers and non-specialist providers (general practitioners) in low-income countries are responsible for a large portion of trauma care.[27] Therefore, human resource development for injury care and appropriate staffing should be prioritised at all levels of healthcare delivery alongside the provision of required infrastructure, equipment and supplies to facilitate the better delivery of trauma care.[27] Models such as Rwanda's Human Resource for Health program should be improved on and strengthened to optimise trauma care specialist training.[28] Training and retention of surgeons and anaesthetists, particularly in rural areas, should be encouraged and incentivised.[29]

Rwanda has one of the most robust health insurance population coverages in sub-Saharan Africa, with community-based health insurance held by over 91% of individuals.[30] This facilitates much needed access to care for those who have insurance. However, it is not yet comprehensive enough to benefit all injured patients.[9 31 32] Participants in both interviews and focus group discussions mentioned barriers related to community health insurance. Multiple insurance-related referral processes, long payment procedures within public health facilities for insurance holders and non-acceptance in private pharmacies contributed to the delays. Additionally, procedures to prove possession of insurance meant that many patients in our study experienced further delays. The dissatisfaction with the procedures for checking insurance has been cited as one of the major barriers to community-based health insurance renewal,[33] despite the fact that 97% of those in the scheme say they benefit from it.[31] For the community health insurance to have a strong positive impact on access to healthcare, those limitations have to be addressed,[32] and more efforts should be engaged toward the development of prehospital and interfacility transportation arrangements, strengthening district hospitals and better supporting referral institutions.[34]

Many participants in this study reported being discharged before they perceived that they were fully recovered. They believed that the early discharge was due to insufficient bed space. Early mobilisation and discharge maybe advantageous for trauma patients to reduce complications of deep vein thrombosis, joint stiffness and depression, among others.[35] However, this early discharge was not complemented by the availability of decentralised rehabilitation services. To be able to return to the treating tertiary facility for physiotherapy or other rehabilitation services for appointments beyond the 30 days allowed by community health insurance, the patients in our study were asked to go through the entire referral process again, without which their insurance payments would not be guaranteed. This involved returning to the initial referral health centre, then to the next district facility that had referred the patient, and finally to the tertiary facility with appointments. This time-consuming procedure was viewed as a barrier to continuing in care.

At the University Teaching Hospital of Kigali, up to 84% of orthopaedic trauma patients (most of whom required postdischarge rehabilitation) have been found to come from facilities in other provinces rather than Kigali. This pattern results in a heightened need for interfacility transfer.[34] Our study suggests that a simplified pathway to follow-up care for injury patients may help reduce losses to follow-up. In addition, options for home-based primary care and other innovative methods may need to be considered and tested,[36] based on the needs and preferences of the communities. These reflections align with other studies. For example, using an online questionnaire distributed through a variety of international and regional organisations from the various WHO regions, Darci et al found that 31.8% of respondents viewed providing rehabilitation services within specialised hospitals and units completely unacceptable, relative to general hospitals or non-specialised units. Up to 79.3% of the same respondents preferred community-based rehabilitation over hospital-based rehabilitation.[37]

This study is not without its limitations. While Kigali is the major conurbation in Rwanda, it is not the only one and our findings from Kigali may not be representative of other urban settings. Likewise, the rural interviews were conducted only in Burera, and certain nuances of injury care access within rural Rwanda may have been missed because of the wide variation in rural settlements while we recognise that barriers may differ slightly by district setting. Additionally, most (75%) of the patients were male, while this may mean that we missed barriers mostly experienced by females; it should be noted that the majority of injured patients in Rwanda are male. In addition, this study did not ensure a range of injury by anatomical/pathological types, severity, mechanism across participants, which may have helped to elicit specific barriers. However, the urban and rural spread of participants was used as a proxy for inclusion of injuries by socioeconomic status. It must also be noted that the three-delay and four-delay frameworks have been critiqued as being too simplistic.[38] The model has been critiqued for being too unidimensional[39] and sequential[40] in contrast to more sophisticated delay models like the access to healthcare conceptual model that provides a richer and more detailed set of factors and interactions.[41 42] The four-delay framework may miss the complex reality of real-life scenarios and is not directly focused on identifying potential solutions.

## CONCLUSION

In contrast to previous studies that focused primarily on the delay to receiving care in health facilities, this qualitative study provides insights into barriers to seeking, reaching, receiving and remaining in care from the injured patients', caregivers' and community leaders' perspectives. In both rural and urban settings, hindrances along the spectrum of care, from injury to rehabilitation, and at various levels of the health system, have been identified. These barriers should be considered and guide injury interventions and health facility organisation. Human resources for health strengthening specific to injury care, facility reorganisation to reduce in-facility waiting times, and revision of the complex follow-up referral system are priority areas for intervention. Future health policy planning should strengthen trauma care by focusing and addressing these barriers.

**Author affiliations**
[1]Single Project Implementation Unit, University of Rwanda, Kigali, Rwanda
[2]Institute of Applied Health Research, University of Birmingham, Birmingham, UK
[3]Center for Equity in Global Surgery, University of Global Health Equity, Kigali, Rwanda
[4]Program in Global Surgery and Social Change, Harvard Medical School, Boston, Massachusetts, USA
[5]Department of Obstetrics and Gynecology, St Olav's Hospital, Trondheim University Hospital, Trondheim, Norway
[6]Malawi-Liverpool-Wellcome Trust Research Institute, Blantyre, Malawi
[7]Institute of Life Course and Medical Sciences, University of Liverpool, Liverpool, UK
[8]Department of Global Health, Centre for Global Surgery, Stellenbosch University, Stellenbosch, South Africa
[9]Medical Research Council/Wits University Rural Public Health and Health Transitions Research Unit, Faculty of Health Sciences, School of Public Health, University of the Witwatersrand, Johannesburg, South Africa
[10]School of Medicine and Pharmacy, University of Rwanda College of Medicine and Health Sciences, Kigali, Rwanda

**Contributors** PN: formal analysis, investigation, data curation, writing - original draft, visualisation, validation, formal analysis, project administration. AI: methodology, formal analysis, writing - review and editing. BTA: formal analysis, validation, writing - original draft, writing - review and editing, visualisation. A-MA-L and MLO: formal analysis, writing - review and editing. JD: conceptualisation, validation, resources, writing - review and editing, supervision, funding acquisition. AB: writing - review and editing, supervision. JCB: conceptualisation, resources, writing - review and editing, supervision, funding acquisition. JCB and JD: Guarantors

**Funding** Funding for this study was provided by the UK National Institute of Health Research (NIHR), award number 130036.

**Disclaimer** The views expressed in this publication are those of the author(s) and not necessarily those of the NIHR or the UK government.

**Competing interests** None declared.

**Patient and public involvement** Patients and/or the public were involved in the design, or conduct, or reporting, or dissemination plans of this research. Refer to the Methods section for further details.

**Patient consent for publication** Not applicable.

**Ethics approval** This study involves human participants and was approved by University of Birmingham Research Ethics Committee, UK (ERN_20-00880) National Health Research Committee for Rwanda (NHRC/2020/PROT/044). Participants gave informed consent to participate in the study before taking part.

**Provenance and peer review** Not commissioned; externally peer reviewed.

**Data availability statement** Data are available upon reasonable request.

**ORCID iDs**
Barnabas Tobi Alayande http://orcid.org/0000-0002-1326-6452
Maria Lisa Odland http://orcid.org/0000-0003-4340-7145
Jean Claude Byiringiro http://orcid.org/0000-0002-6445-1797

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
