## [Reviewer comments · BMJ Open]

ARTICLE DETAILS

TITLE (PROVISIONAL)	Barriers to equitable access to quality trauma care in Rwanda: a qualitative study
AUTHORS	Nzasabimana, Pascal; Ignatowicz, Agnieszka; Alayande, Barnabas; Abdul-Latif, Abdul-Malik; Odland, Maria Lisa; Davies, Justine; Bekele, Abebe; BYIRINGIRO, JEAN CLAUDE

VERSION 1 – REVIEW

REVIEWER	Whitaker, John King's College London Faculty of Life Sciences and Medicine, Centre for Global Health and Health Partnerships
REVIEW RETURNED	19-May-2023

GENERAL COMMENTS	Overall I enjoyed the article which provides a valuable patient and community perspective on important barriers to injury care throughout the patient journey. I've made some comments which are hopefully useful to strengthen the manuscript. Introduction: Road traffic crash vs collision – consider most suitable terminology Timely trauma care also needs to be good quality – consider including this dimension (line 40) Does injury patient perspectives here refer specifically to in Rwanda? May be worth clarifying as has been explored in other contexts. (p6 line 35) Methods: Consider describing that SQRQ was used as a guide to the study report which enhances the rigour. Can you reference the Ghana and South Africa equivalent studies? I think more detail is needed on how the interview and focus groups were conducted, what questions were used and why and how the participants were orientated to the study including definitions of injury and delays etc. Were the study guides in English or local language? I think a dedicated study context paragraph is required (as per SQRQ) to aid the reader with understanding of the injury care pathways, terrain, financing etc.
---

	Carers are mentioned as potential participants in methods. Are carers included in the hospital register, or were they identified via the trauma patient? Were any actual participants, not seen in results? Page 8 line 31 – do you mean access rather than assess? This criteria would exclude those who have successfully remained in care and could introduce bias – perhaps discuss. Did you consider ensuring a range of injury anatomical/pathological types, severity, mechanism and SES across participants all of which may elicit specific barriers. Regarding definitions of delay to receiving care, definitive management may appropriately take a while for some injuries that require repeat theatre attendances, whereas others required no intervention. This may lead to false impression of delay for appropriate injury management in some cases. Perhaps clarify what is meant as definitive or justify why not damage control intervention. Results: Head injury patients participated - Did you consider whether a head injury patient might have cognitive deficit inhibiting participation? Was this an exclusion criteria as well? Can you report mechanism of injury as well and were these distributed well across groups? The mechanism may have implications for particular barriers worth exploring, e.g. assault vs RTC vs fall. You report a statistical test for urban vs rural participant characteristics - what test was done and is it really meaningful for such small numbers? Perhaps mention in methods too. You report barriers which were the most mentioned. As you've quantized the qualitative data in some way can you show that? How about a word cloud that might pictorially represent this (e.g. for barriers mentioned more than a certain threshold if too many to neatly display). Discussion: Perhaps some limitations of the 4 delays approach to identifying potential solutions could be mentioned. Ethics: What about community leadership approval for the community dimension of the project? SQRQ: Not sure the context description is well developed. Could definitely be strengthened with pertinent information regarding social, geographical and health facility information pertinent to the study. More needed on the instrument, how it was developed, how the questions were structured, the interviews conducted etc. What was meant by injury?
--	--

REVIEWER	Makkink, Andrew University of Johannesburg, Emergency Medical Care
-----------------	---

REVIEW RETURNED	15-Jun-2023
GENERAL COMMENTS	Thank you for an interesting study that addresses an important aspect of healthcare, delays in healthcare, specifically within a resource-constrained environment. This was a well-written manuscript and I have some minor comments. How were the barriers in Table 1 identified for inclusion? Was it a single mention, or did the authors have some sort of threshold? Perhaps just a sentence or two that provides more information for the reader to be able to determine why and how each barrier made it into the table. The table could do with some formatting- there seems to be an empty column to the left which could be removed. I would also suggest splitting the table into the four major themes- it is really long the way it is currently presented. Perhaps consider removing the horizontal lines between the delays and rather grouping themes in cells with bullet point markers. This may make it easier for the reader, it would for me. Please check in-text referencing where multiple reference are grouped together for correctness in formatting. Overall an interesting and well-structured read. Thank you.

VERSION 1 – AUTHOR RESPONSE

Reviewer 1

4. Introduction: Road traffic crash vs collision – consider most suitable terminology

The most suitable terminology is road traffic collisions, and this has been applied throughout the manuscript.

5. Timely trauma care also needs to be good quality – consider including this dimension (line 40)

The dimension of quality care has been introduced.

6. Do injury patient perspectives here refer specifically to in Rwanda? May be worth clarifying as has been explored in other contexts. (p6 line 35)

This has been clarified. “However, the perspectives of the injured patients “in the Rwandan context” have not yet been explored.”

7. Methods: Consider describing that SQRQ was used as a guide to the study report which enhances the rigour.

Under methods, we have included the statement that “The Standards for Reporting Qualitative Research (SRQR) checklist guided all components of the writing and reporting of this study.”

8. Can you reference the Ghana and South Africa equivalent studies?

This study was conducted concurrently with studies in Ghana and South Africa as part of the Equitable access to quality trauma systems in Lower and Middle-Income Countries: A qualitative study assessing gaps and developing priorities, and the available study has been referenced.

9. I think more detail is needed on how the interview and focus groups were conducted, what questions were used and why and how the participants were orientated to the study including definitions of injury and delays etc. Were the study guides in English or local language?

The data source sub-chapter has been fragmented to distinctively highlight the Context, Sampling strategy, Data collection methods, Data collection instruments and analysis, to better address these comments.

More detail on how the interview and focus groups were conducted has been included under the new headings 'sampling methods', 'data collection methods', and 'data collection instruments and analysis.' We have also included the focus group discussion and individual interview guides as supplementary files.

Under data collection methods:

"Participants were oriented to the study by the interviewer (PN) who explained the aims and objectives of the study to participants in Kinyarwanda, ensuring verbal feedback from potential participants to confirm that they understood the study, and their potential role. Written informed consent was subsequently obtained. Individual interviews lasted 30 minutes on average, and focus groups discussions lasted between one and one and a half hours for six to nine participants."

Under data collection instruments and analysis:

"Data collection was guided by study instruments designed by qualitative experts in English, which were translated to Kinyarwanda."

10. I think a dedicated study context paragraph is required (as per SQRQ) to aid the reader with understanding of the injury care pathways, terrain, financing etc.

A dedicated context paragraph has been introduced describing the study location, the geographical context, injury care pathways, terrain, financing etc. (See 'Context' under 'Methods')

"The study was conducted in Rwanda's rural Burera District, located in the Northern Province, and in Kigali City, the country's urban capital. Kigali City has a population of approximately 1,135,428 people concentrated on 1,760/km², whereas Burera district has a population of 336,455 people distributed over 522/km²(10). Burera, in terms of population and infrastructure, is a typical rural Rwandan district. Rwanda's urbanization is dominated by Kigali, which houses nearly 60% of the urban population and continues to experience rapid population and economic growth while modernizing and upgrading its infrastructure and transportation systems(11). Road traffic injuries are a major cause of trauma deaths in Rwanda, particularly among young men (12). Injury care pathways generally start with community health workers who are embedded in the communities, who refer to local health centers, and subsequently to second tier district hospitals, who refer to tertiary teaching hospitals and eventually to the country's single quaternary health institution. Many health facilities have ambulances that can transport injured patients to higher levels of care. The country's terrain is largely hilly. Most Rwandans (over 90%) are covered by Mutuelle de Santé, Rwanda's elaborate Community-Based Health Insurance program."

12. Carers are mentioned as potential participants in methods. Are carers included in the hospital register, or were they identified via the trauma patient? Were any actually participants.

Patient caregivers (carers) were identified via the trauma patients. Carers are not included in the hospital register. They also took part in the study, and carer/caregivers actually participated in the interviews. "Thirty-six (71%) of the participants were patients (service users), while the rest were caregivers and community leaders." (Under Results)

13. Page 8 line 31 – do you mean access rather than assess? This criteria would exclude those who have successfully remained in care and could introduce bias – perhaps discuss.

Thank you- accessing has been used to replace assessing. Criteria has been rephrased for clarity. “Individuals under the age of 18 or with any intellectual impairments that would limit informed consent and participation, as well as service users accessing acute care during the study period, were excluded. Individuals who returned for follow up and successfully remained in care however were not excluded.” For instance, individuals who had returned for follow up or rehabilitation were included.

14. Did you consider ensuring a range of injury anatomical/pathological types, severity, mechanism, and SES across participants all of which may elicit specific barriers.

Thank you for this important observation. We considered ensuring a range of injury by anatomic/pathologic type, severity, or mechanism, as well as socio-demographic characteristics, which could have helped elicit specific barriers, however, potentially this would lead to an unrealistic sample, as some mechanisms/types of injury (and their barriers) may be quite rare. This has been included in the limitations. The benefit of our approach is that we effectively ended up with participants that represent the majorities of injuries.

“In addition, this study did not consider ensuring a range of injury by anatomical/pathological types, severity, mechanism across participants, which may have helped to elicit specific barriers. However, this may have also biased the findings towards rarer barriers that are not experienced by the majority of injured people. The urban and rural spread of participants was used as a proxy for inclusion of injuries by socioeconomic status.”

14. Regarding definitions of delay to receiving care, definitive management may appropriately take a while for some injuries that require repeat theatre attendances, whereas others required no intervention. This may lead to false impression of delay for appropriate injury management in some cases. Perhaps clarify what is meant as definitive or justify why not damage control intervention.

The definition of the third delay has been clarified. “The third delay (delay in receiving care) was defined as delays beginning from the point of entering a capable facility to the point of receiving specific management for the injury. This could include damage control interventions for the injury initiated upon admission to a hospital intended to manage an injured patient’s condition or injury. This specific management can be operative surgical care when there is the indication for surgery, but is often non-operative, like the application of casts, administration of medications, or admission for observation.”

15. Results: Head injury patients participated - Did you consider whether a head injury patient might have cognitive deficit inhibiting participation? Was this an exclusion criteria as well?

These previous head injured patients were fully recovered patients at the time of their interviews. Exclusion of cognitively impaired or unrecovered head injured patients was in the exclusion criteria and has been included in the manuscript. Any patients with cognitive deficit that would limit informed consent and participation were excluded. See under Sampling strategy. “Individuals...with any...cognitive deficit that would limit informed consent and participation...were excluded.”

16. Can you report mechanism of injury as well and were these distributed well across groups? The mechanism may have implications for particular barriers worth exploring, e.g. assault vs RTC vs fall.

Mechanism of injury and its distribution has been described in the results. Falls from heights were more in the hilly rural settings (40%) compared to urban Kigali (10%). Only 1 patient presented as a

victim of physical assault. “The major mechanism of injury was involvement in road traffic collisions in the urban setting (9, 90%), and less so in the rural settings (5, 50%). There were more falls from heights in the hilly rural settings (4, 40%), than in the urban areas. Only 1 rural report of assault was captured.

17. You report a statistical test for urban vs rural participant characteristics - what test was done and is it really meaningful for such small numbers? Perhaps mention in methods too.

In the previous draft, Fischer’s exact tests were used to compare urban versus rural participant characteristics- specifically, the type of injury. “There was no statistically significant difference in the patient’s type of injury by rural or urban location ($X^2 = 8.444$, $df = 5$, $p = 0.133$).” We have removed this from the manuscript as it is a qualitative paper, and as the reviewer has pointed out, without a power calculation, the results are not meaningful for the total number.

18. You report barriers which were the most mentioned. As you’ve quantitized the qualitative data in some way can you show that? How about a word cloud that might pictorially represent this (e.g. for barriers mentioned more than a certain threshold if too many to neatly display).

We have quantized the qualitative data and have included a word cloud that pictorially represents this. Figure 1.

19. Discussion: Perhaps some limitations of the 4 delays approach to identifying potential solutions could be mentioned.

Limitations of the 4-delay approach have been acknowledged in the discussion. “It must also be noted that the 3- and 4-delay frameworks have been critiqued as being too simplistic (Bohren et al, 2014). The model has been critiqued for being too unidimensional (Sorensen et al, 2011) and sequential (D’Ambruso et al, 2010) in contrast to more sophisticated delay models like the access to healthcare conceptual model that provides a richer and more detailed set of factors and interactions (Combs Thorsen et al, 2012) (Khan & Bhardwaj, 1994). The 4-delay framework may miss the complex reality of real-life scenarios and is not directly focused on identifying potential solutions.”

20. Ethics: What about community leadership approval for the community dimension of the project?

Community leaders within the facilities’ catchment area also gave permissions and participated in the interviews. This has been included under ethics.

21. SQRQ:

Not sure the context description is well developed. Could definitely be strengthened with pertinent information regarding social, geographical and health facility information pertinent to the study.

Context description has been better developed with information pertinent to the study. See Context.

22. More needed on the instrument, how it was developed, how the questions were structured, the interviews conducted etc. What was meant by injury?

More information on the instrument and data collection methods have been added under the newly introduced sections “Data collection methods” and “Data collection instruments and analysis”. Data collection was guided by study instruments designed by qualitative experts in English, which were translated to Kinyarwanda. All the instruments have been included in Supplementary Information.

Definition of injury included. "We defined injury as wound or a condition of the body caused by external force or exchange of energy between the body and the environment of such magnitude that is beyond the resilience of the body."

Reviewer 2

1. How were the barriers in Table 1 identified for inclusion? Was it a single mention, or did the authors have some sort of threshold? Perhaps just a sentence or two that provides more information for the reader to be able to determine why and how each barrier made it into the table.

To identify these barriers in table 1, the following process described under 'data collection instruments and analysis' was used: "...all transcripts of interviews and focus group discussions were coded in parallel by a context and local language fluent member (PN) and qualitative data experts (AI and MLO). The analysis team met on a regular basis to discuss the coding process. Any conflicts that arose during the independent coding process were resolved by group consensus. The analysis procedure was the same for interviews and focus groups. Following coding and the identification of initial categories, data from interviews and focus groups were combined. The final list of themes was reviewed and agreed upon by the entire investigator team."

We used at least single mention to include these barriers.

2. The table could do with some formatting- there seems to be an empty column to the left which could be removed. I would also suggest splitting the table into the four major themes- it is really long the way it is currently presented. Perhaps consider removing the horizontal lines between the delays and rather grouping themes in cells with bullet point markers. This may make it easier for the reader, it would for me.

Table re-formatted as per reviewer's comments

Additional comments from editor on resubmission- 1. Required Supplementary format:- Please re-upload your Supplementary files in PDF format. This has been done.

2. Figure 2 consent:- Your article has been flagged during editorial processing as it includes potentially identifiable patient information and/or reports on patients and a signed BMJ consent form has not been uploaded.

A BMJ consent form signed by the provider of the image in Figure 2 has been included. The current Figure 2 does not contain any patient information or reports on patients. No patient faces or patient materials are photographed. However a signed BMJ consent form has been uploaded. we have also included a BMJ anonymization checklist to show that there is no patient identifiable information on the the image.

VERSION 2 – REVIEW

REVIEWER	Whitaker, John King's College London Faculty of Life Sciences and Medicine, Centre for Global Health and Health Partnerships
REVIEW RETURNED	08-Aug-2023

GENERAL COMMENTS	I think my comments have been adequately addressed. The word cloud is a helpful addition. I am happy to recommend publication.
--